Mam-Incept-Net: a novel inception model for precise interpretation of mammography images

http://orcid.org/0000-0002-9219-3203 Tandirovic Gursel Amira 1 agursel@atu.edu.tr
http://orcid.org/0000-0002-9074-0189 Kaya Yasin 2 ykaya@atu.edu.tr
1 Department of Electrical and Electronics Engineering, Adana Alparslan Turkes Science and Technology University , Adana , Turkey
2 Department of Artificial Intelligence Engineering, Adana Alparslan Turkes Science and Technology University , Adana , Turkey
Fasi Massimiliano
Electronic publication date: 2025 Aug 28
Publication date: 2025
Volume: 11
Electronic Location ID: e3149
Received 2025 Mar 28; Accepted 2025 Jul 31
Copyright: © 2025 Tandirovic Gursel and Kaya
Copyright year: 2025
Copyright holder: Tandirovic Gursel and Kaya
License: This is an open access article distributed under the terms of the Creative Commons Attribution License, which permits unrestricted use, distribution, reproduction and adaptation in any medium and for any purpose provided that it is properly attributed. For attribution, the original author(s), title, publication source (PeerJ Computer Science) and either DOI or URL of the article must be cited.
License URL: https://creativecommons.org/licenses/by/4.0/

Keywords: Breast cancer, Mammogram, BI-RADS categories, Deep transfer learning, Early diagnosis, Deep learning

Funding: The authors received no funding for this work.

==============================
Early diagnosis of breast cancer through periodic screening is a vital ally in the fight for survival. Mammography, recognized as one of the most widely used and cost-effective tools for detecting early signs of asymmetry, calcification, masses, and architectural distortion in breast tissue, plays a significant role in nearly all screening scenarios. However, the interpretation and scoring of mammograms is a complex multi-parameter process that frequently leads to false-positive and false-negative results. This article introduces a new deep-learning-based model that classifies mammograms according to the Breast Imaging Reporting and Data System (BI-RADS) assessment categories. The model is trained on a private dataset, intentionally excluding no BI-RADS categories. A novel deep neural network architecture is employed to more accurately classify breasts, including their boundaries, as regions of interest (ROIs). The ConvNeXt architecture serves as a feature extractor for lower-level features, which are then combined with the layers of a randomly initialized naive inception module to capture higher-level features. Diagnosis is achieved through three experimental tests, yielding accuracy rates ranging from 82.08% to 86.27%. These promising accuracy levels, in comparison to previous studies, can be attributed to a more comprehensive approach to addressing BI-RADS scoring challenges. In addition to pursuing further enhancements in accuracy, future research should consider integrating prior radiology reports to create a more realistic end-to-end computer-aided detection system.

Introduction

Breast cancer (BC) is a generic term for a group of biologically and molecularly heterogeneous diseases starting when abnormal cells of the breast show uncontrollable growth and exceed their usual boundaries. Driven by genetic and epigenetic alterations in the mechanisms controlling the proliferation, survival, and migration of breast cells, the disorder manifests as the invasion or spreading of adjacent cells to other organs (Hong & Xu, 2022).

According to the GLOBOCAN 2020 database, which was founded by the World Health Organization (WHO) and reports data for 36 cancer types in 185 countries, BC is the most common cancer worldwide, with prevalence accounting for about 12.5% of all new annual cases regardless of gender. It is to be noted that BC alone accounts for 30% of female cancers (Siegel, Giaquinto & Jemal, 2024, Siegel et al., 2022). Moreover, according to the reports of the American Cancer Society (ACS), approximately 13% of the world’s female population is expected to be diagnosed with invasive BC in their lifetime, while a man’s lifetime risk is about 0.12%. Although the causes are not fully understood, and BC is typically considered to be mainly sporadic, several risk factors are already known to affect the likelihood of its development, including gender, age, family and personal history, and certain lifestyle factors (Sung et al., 2021).

Overall, for several reasons, both cancer incidence and mortality rates are growing globally, and BC is no exception. In the study (Arnold et al., 2022), it is pointed out that by 2020, BC is projected to grow to about three million newly diagnosed cases and one million deaths per year. Fortunately, the last annual report points out that despite the positive correlation, the gap between incidence and mortality rates grows in favor of incidence mainly because the relative survival rate has increased substantially during several decades (Siegel et al., 2022).

As the cancer stage is the most important determinant of outcome, experts emphasize that early detection is a cornerstone in the fight for the survival of BC patients (Shah et al., 2024a, Shah, Ullah Khan & Abrar, 2024). The positive effects on the disease outcome provide a strong basis for the practice of scanning, whose primary goal is to identify clinically occult tumors that are likely to become fatal before the cells begin to metastasize. Despite much debate, primarily focused on the risks, benefits, and starting age, it is generally accepted that annual mammograms are the best means of screening healthy women for breast cancer (Gilbert, 2020). The radiology community recognizes screening mammography as the gold standard in the early detection of BC and a cost-effective modality that could catch most of the changes likely to mutate in the next few years (American Cancer Society, 2020). However, factors such as poor image quality, breast density, as well as errors in image analysis and interpretation frequently lead to overdiagnosis or more serious misdiagnosis or missed diagnosis, which, in turn, radically change treatment patterns or, even worse, restrict the alternative treatment options.

Despite state-of-the-art digital mammography systems providing all the easier control over the image parameters, mistakes related to poor mammography image quality, primarily influenced by the mammogram unit itself, are not rare. Additionally, as pointed out in Elmore et al. (2016), analyzing and interpreting mammograms, typically performed by radiologists or nuclear medicine physicians, is a prolonged and tedious task (Shah et al., 2024b). Additionally, it is a skill that requires effective practice, attention, experience, and training. Several strategies, including obtaining two views per breast, comparing them with prior mammograms, and double reading, have been implemented to improve screening performance. Double reading, whether independently or in conjunction, is widely employed in Europe, often supplemented by conventional or state-of-the-art computer-aided detection systems (CADs). Still, it is not a standard practice in the United States, primarily due to additional costs and overloading of the radiologists. Instead, CADs are proposed as the concept of a second opinion (Vobugari et al., 2022).

The automatic detection and classification of breast lesions by CADs is a topic that emerged in the 1990s. Primarily, due to their low specificity and time-consuming data processing, they had little to no impact on radiologists’ performance (Kohli & Jha, 2018). Hence, they soon gave way to machine learning (ML)-based classifiers. Deep learning (DL) algorithms are a type of ML family whose strong point is the enviable handling of complex, nonlinear functions characteristic of real-world problems, such as mammogram classifiers (Jiménez-Gaona et al., 2024; Raiaan et al., 2024; Jakkaladiki & Maly, 2024).

More specifically, Sabani et al. (2022) employed a deep convolutional neural network (CNN) for Breast Imaging Reporting and Data System (BI-RADS)-based classification of mammographic images and indicated that CNN models can operate for the automatic detection and standardized, observer-independent classification of mammograms, regardless of the presence of microcalcifications. In another study, which utilized BI-RADS categorization using CNNs, was presented by Tan et al. (2025) and achieved promising results. Ahmed & Nandi (2024) proposed an EfficientNet architecture for breast cancer diagnosis from mammograms. In another study (Petrini et al., 2022), which employed the EfficientNet architecture, the authors utilized a single-view whole-image classifier.

The primary objective of this study is to develop a robust and intelligent CAD system that improves the accuracy and reliability of breast cancer detection from digital mammograms. Specifically, we aim to classify mammographic images into four clinically meaningful categories: BI-RADS 0, BI-RADS 1, benign, and malignant, by leveraging deep CNNs enriched with Inception and ConvNeXt modules. The proposed solution is specifically designed to support radiologists by reducing diagnostic errors, enhancing screening efficiency, and facilitating faster interpretation, particularly in regions where experienced radiologists are scarce.

To effectively address this goal, it is essential to understand the clinical framework and diagnostic criteria underlying mammogram interpretation, particularly the BI-RADS categorization system used by radiologists. There is no way to genuinely depict the engineering problem of mammogram reading without the basic knowledge about BC-related abnormalities that should be looked for and how they are reported. Unless otherwise specified, a screening mammogram consists of four standard views: mediolateral-oblique (MLO) and cranio-caudal (CC) views per breast. The screening mammogram is carefully examined and compared to the prior one according to six criteria: skin lesions, density, bilateral symmetry, architectural distortion, masses, calcification, and lymphadenopathy (Spak et al., 2017). The mammogram reading has been standardized since 1976. Since ACS recommended standardization, radiologists and healthcare providers have used a standard mammogram reading system, the Breast Imaging-Reporting and Data System (BI-RADS). Table S1 summarises BI-RADS assessment categories. It is worth noting that the same system is also used for reading breast ultrasounds or breast magnetic resonance imaging (MRI) exams.

This study proposes a new BC detection approach for classifying digital mammogram images into four categories: BI-RADS 0, BI-RADS 1, benign, and malignant. Apart from introducing all the BI-RADS categories, the model deals with all sub-problems considered as parameters for BI-RADS scoring simultaneously. From the engineering point of view, the study provides several important contributions, the main of which are listed as follows: This work introduces a novel fusion model for BC classification that combines a ConvNeXt-based CNN and Inception blocks.

The study also presents a new pre-processing step for automatically detecting left flipping in breast images.

We evaluated the proposed model with three test scenarios on a private dataset, and promising results were attained.

The rest of the article is organized as follows: In ‘Related Works’, we discuss related works, first addressing the issue from a medical perspective and then analyzing several existing DL-based works. Details of the proposed DL-based approach are represented in ‘Material and Methods’, while obtained results are presented and discussed in ‘Results’. ‘Discussion’ discusses the study in relation to other studies in the literature. Finally, ‘Conclusion’ summarizes the article and provides an overview of the challenges and future research directions.

Related works

Conventional CAD systems are mostly focused on reducing the number of false-positive findings. They rely on human-designed (hand-engineered) features and usually examine just one or two of several determiners, such as lesions and calcifications. The determiner(s) is/are examined for two, in essence, separate tasks: detection and diagnosis. The detection process begins with identifying potential lesions in the fibro glandular tissue and culminates in the classification of detected regions (Sechopoulos & Mann, 2020). The process of detecting the determiner(s) is primarily based on some intermediate steps, such as segmentation and feature extraction, regardless of whether it utilizes region of interest (ROI) or the whole image. The extracted features are used to categorize the determiner under consideration. However, shortly after their introduction to the market, medical practitioners warned against the low specificity and sensitivity of CADs (Abdelhafiz et al., 2019).

They are reported to have missed some types of cancers, such as non-calcified lesions (Chan, Samala & Hadjiiski, 2019). The main reason for failure is the complexity of the task. In mammogram images, cancer is presented through architectural distortions, masses, and micro-calcifications (Sampat, Markey & Bovik, 2005), whose variations in contrast appearance, size, shape, texture, or clustering pattern obstruct either detection itself or classification, which compels designers to tackle those sub-problems independently of each other. Hence, it seems that CADs are left behind mainly due to the sheer variety of data to be processed, which is a versatile field for ML and DL algorithms.

The rapid progress in image processing, computer technology, and ML and DL algorithms, made in recent years, with a remarkable impact on performance concerning pattern recognition and classification accuracy, has opened up an opportunity to overcome the system limitations of conventional CADs. Accordingly, recent studies on this topic can be collected under four headings: improvements related to pre-processing, segmentation, feature extraction, and/or classification accuracy. In studies on mammography, DCNN is the commonly used method for pattern recognition. It is widely utilized for detecting and classifying either micro-calcifications or masses. CNN models were first used in mammography diagnosis as a true-false classifier of masses in 1994 (Chan et al., 1994). Not long after, a similar model was applied to micro-calcifications by Zhang & Yeung (2012). Despite many shortcomings, such as the limited number of convolutional layers and kernels in each layer, which are crucial for the learning process, these early CNNs played a pivotal role in advancing CNNs for pattern recognition in medical imaging (Chan, Hadjiiski & Samala, 2020).

Another important model, proposed in Al-antari et al. (2018), is a fully integrated CAD system for detecting masses tested on the DMSM dataset. The proposed systems utilize a regional DL approach, You Only Live Once (YOLO) for detection, and the full resolution convolutional networks (FrCN) model for segmentation. Finally, the obtained data is recognized and classified into benign and malignant by the ConvNet deep convolutional neural network (DCNN) model. The proposed system is characteristic of high-speed testing of only 12.23 s per image and an accuracy of 95.64%. In another study (Baccouche et al., 2021), a YOLO-based end-to-end system is proposed to classify breast lesions as calcifications, masses, and tumors without requiring lesion segmentation or malignancy prediction. Based on the fusion model approach, the third improved version, YOLO-V3, is used to localize suspicious lesions within the image before they are fed into a CNN classifier to generate a convolutional feature map for classification. The model is performed not only on two public datasets, CBIS-DDSM and INbreast, but also on an independent private dataset. The best overall detection accuracy rate of about 92% is achieved for the private dataset. Although the proposed model is designed to detect and classify both mass and calcification lesions, it generally performs better for mass lesions, with a maximum detection accuracy rate of 96.2% and a maximum inference time of 0.58 s.

In Jiang et al. (2022), the authors proposed an automatic detection and classification of both mass and calcification breast lesions. A priority-aware algorithm (PAA)-based three-stage DL model is employed to robustly locate lesions from the whole mammogram image before combining the ROI classifier and the image classifier, classifying them into malignant and benign categories. The training mammography dataset is formed by integrating three public datasets: MIAS, CBIS-DDSM, and INbreast. One of the interesting double-task studies is the study conducted in Ma & Peng (2022), which presents a novel DL framework scheme for the detection and classification of masses with simultaneous analysis of bilateral symmetry, achieving an accuracy of 86%. Bilateral symmetry is trained using a shared CNN model, while a soft-label-based classifier categorizes masses as normal, benign, or malignant. Experiments are performed on the DDSM dataset.

Unlike the vast majority of research articles on classifying breast lesions as malignant or benign, the work conducted in Heenaye-Mamode Khan et al. (2021) focuses on detecting and classifying four main types of breast abnormalities simultaneously using a DCNN modality. With a performance of 88%, the CBIS-DDSM public dataset is pre-trained by ResNet50, while training is provided via the enhanced CNN model developed by the authors. Although the model should be improved in terms of accuracy, it could be a crucial milestone for better determining the most appropriate treatment (Mahoro & Akhloufi, 2022).

One of the latest studies proposes a hybrid model in Elkorany & Elsharkawy (2023), where three different DCNN models, Inception-V3, ResNet50, and AlexNet, are utilized as feature extractors. The Term Variance (TV) feature selection algorithm extracts the appropriate features from each CNN model. The TV-selected features from each CNN model are combined to obtain more distinct features before sending them to the multiclass support vector machine (MSVM) classifier. According to the study, the proposed diagnosis model employing three class cases, normal, benign, and malignant, is tested on 322 images of the MIAS dataset and provides 98% classification accuracy (CA) for 90% of the training.

In another recent study (Jiménez-Gaona et al., 2024), the Gan-based data augmentation method was employed to improve breast mammography mass classification, achieving 80.9% accuracy for benign masses and 76.9% accuracy for malignant masses. A lightweight CNN for diagnosing breast cancer from mammography images was proposed by Raiaan et al. (2024), attaining promising results. Teoh et al. (2024) proposed an optimized ensemble deep learning framework for early breast cancer diagnosis through automated microcalcification detection. They employed widespread pre-trained models and achieved promising results. In another study that uses deep ensemble learning, Das et al. (2021) proposed a breast cancer detection model employing an ensemble CNN-based deep learning method. They used raw and decomposed images generated by 1D Empirical Wavelet Transform and attained 98.08% classification success.

Dave et al. (2025) examined studies to highlight the integration of AI into mammography to improve breast cancer diagnosis. The authors conclude that AI has the potential to be transformative in breast cancer screening. In another review study (Shifa et al., 2025), the authors focused on explainable artificial intelligence methods (XAI) in mammography for breast cancer screening. They concluded that XAI helps validate AI results, supports clinician education, and addresses ethical concerns related to black box models. Manigrasso et al. (2025) experimentally analyzed transformers, CNN, and graph-based architectures for breast cancer classification.

It can be said that a great many research articles have been published on the detection and diagnostic modalities of BC in the last decade. Refers to review articles on the topic, there are several articles, such as those conducted in Chougrad, Zouaki & Alheyane (2018), Oza et al. (2022b) and Tariq et al. (2021), which provide a fairly comprehensive analysis of not only the state-of-the-art DL-based BC diagnosis and detection methodologies, but also valuable information on pre-processing modalities, classification types, and datasets. However, few reviews provide any systematic research into a grouping of BI-RADS categories during the classification process.

Table 1 lists 16 additional studies conducted since 2016, 1 year after the first research on DL models was conducted (Dhungel, Carneiro & Bradley, 2015). Apart from classification and grouping according to BI-RADS categories, the table summarizes information regarding datasets and classification algorithms predominantly carried out in the studies. It can be inferred that research to date has mainly focused on binary classification to distinguish between benign and malignant mammogram images with manually marked ROIs by radiologists, which is a crucial drawback to the widespread application of these models. Further analysis reveals a need for a more comprehensive classification method that overlaps with the 7-category BI-RADS assessment model, as shown in Table S1.

Table 1 Summary of previous studies.

Authors (Year)	Dataset/Availability	Classification algorithm	Classification type	
Qian et al. (2024)	INbreast/Moreira et al. (2012)	Multi-feature fusion neural network (MFNet)	Benign group (BI-RADS 1, 2, and 3), malign group (BI-RADS 4, 5, and 6), BI-RADS 2 and 3 are discarded	
Achak & Hedyehzadeh (2023)	CDD-CESM/Khaled et al. (2022)	Resnet-50, Resnet-18, and Densenet-201; the K-Fold10 technique	Benign and malignant non-mass enhancement (NME) lesions	
He et al. (2023)	CBIS-DDSM/Lee et al. (2017)	Semantic Pyramid Network with a Transformer Self-attention (SPN-TS)	Detection/Benign and malign	
Lou et al. (2022)	INbreast/Moreira et al. (2012)	Deep CNN	Benign group (BI-RADS 1 and 2), malign group (BI-RADS 4, 5, and 6), BI-RADS 2 and 3 are discarded	
Walton, Kim & Mullen (2022)	CBIS-DDSM/Lee et al. (2017)	CNN	Co-locating lesions, classification according to the size of the mass type lesions	
Malebary & Hashmi (2021)	DDSM + MIAS/Heath et al. (1998), Suckling et al. (2015)	ResNet, LongShortTermMemory of RecurrentNeuralNetwork(RNN-LSTM) + ResNet-ResNetnetwork + ResNet-VGG	Normal (BI-RADS 1), benign (BI-RADS 2 and 3), malign (BI-RADS 4, 5, 6)	
Aly et al. (2021)	INbreat/Moreira et al. (2012)	YOLO-V3; ResNet and InceptionV3	Mass classification, benign (BI-RADS 2 and 3), malign (BI-RADS 4, 5, and 6)	
Cai et al. (2020)	Nanfang Hospital (NFH), Guangzhou/Cai et al. (2019)	Deep learning method using neutrosophic boosting	Grading micro-calcification clustering: Three group classification, BI-RADS 3, 4, and 5	
Al-antari, Han & Kim (2020)	DDSM + INbreast/Heath et al. (1998), Moreira et al. (2012)	YOLO detector, modified InceptionResNet-V2 classifier	Clasificaton of breast lesions (Benign and malign)	
Lu, Loh & Huang (2019)	Local dataset	Fully cannected CNN	Binary (Benign and malign)	
Gandomkar et al. (2019)	Local dataset	Convolutional Neural Networks (CNN), Visual Analogue Scales (VAS)Inception-V3 pre-trained on the ImageNet	The low-risk group (BI-RADS 1 and 2), the high-risk group (BI-RADS 3 and 4)	
Kim et al. (2018)	Yonsel University health system/In-house dataset	ResNet-5 0	Sorting out cases (BI-RADS 1 cases and others)	
Jung et al. (2018)	INbreast + GURO/Moreira et al. (2012), In-house dataset	RetinaNet	A mass detection based model, malignancy Binary (Benign and malignant)	
Dhungel, Carneiro & Bradley (2017)	INbreat/Moreira et al. (2012)	Deep learning, Bayesian optimization, Transfer learning	No mass (BI-RADS 1), benign mass (BI-RADS 2 and 3), malignant mass (BI-RADS 4, 5, and 6)	
Geras et al. (2017)	Health Insurance Portability and Accountability (HIPAA)	Multi-ViewDeepConvolutionalNeuralNetwork	Incomplete (BI-RADS 0), negative (BI-RADS 1), benign mass (BI-RADS 2)	
Arevalo et al. (2016)	BCDR/(Lopez et al., 2012)	Various CNN algorithms: CNN3, CNN2, HGD, HOG, DeCAF, and Hcfeats	Mass malignancy Binary (Benign and malign)	
Lévy & Jain (2016)	DDSM/(Heath et al., 1998)	ShallowCNN (the baseline model), AlexNet, and GoogLeNet	Binary (Benign and malign) BI-RADS 0, 1, 2, 3 as benign and BI-RADS 4,5	

A closer look into the problem reveals several critical gaps in this field, including the need for more widely accepted pre-processing techniques, the lack of multi-task models capable of detecting multiple abnormalities, and the requirement for more balanced, comprehensive, and well-labeled datasets. Additionally, almost no attention has been paid to the BI-RADS 0 category, which specifically indicates that there is insufficient information to allow for a complete evaluation. That is to say, the 0 category points to unclear findings covering two analogous but different diagnostic settings: suspicious findings, such as oval or round lesions with circumscribed or partially obscured margins and focal asymmetry, which need additional imaging to be defined, or absence of results from previous check-ups to be compared with. In the retrospective study conducted in Zanello et al. (2011), which assessed the sonography outcome of patients with mammograms initially classified as BI-RADS 0 category, it was noted that 29.8% of the patients had sonogram results classified as BI-RADS 4, while cancer was found in 4.1% of the testing group. Contrary to several studies, such as those conducted in Qian et al. (2024), Malebary & Hashmi (2021), or Le et al. (2024), the results indicate that this category can be neither discarded nor integrated with other categories.

Materials and Methods

This section focuses on particularizing the dataset and the proposed hybrid deep learning model, which consists of the following sub-steps: pre-processing, deep feature extraction, and classification, as illustrated in Fig. 1.

Figure 1 (A–C) A block diagram of Mam-Incept-NET for mammogram classification.

Dataset

The raw dataset comprises mammograms of 802 patients, collected over time in two large hospitals in Adana, Turkey. In a standard mammogram examination, a mammogram is a set of four images, one MLO and one CC examination per breast. Instead of 3,208 images, the dataset comprises 2,566 images captured by two mammography machines from different vendors. Missed mammograms occurred because some patients were examined during post-operative follow-up after mastectomy or ruled out due to poor quality, and can be determined easily by following the image labels. Additional details about classes are given in Table 2.

Table 2 Dataset details and image distribution for experimental tests.

Dataset	Experimental tests	
Category	Size	Test 1	
BI-RADS 0	402	BI-RADS 1	Other	
BI-RADS 1	751	751	1,815	
BI-RADS 2	553	Test 2	
BI-RADS 3	439	BI-RADS 0	Other	
BI-RADS 4, 5, and 6	421	402	2,164	
Total	2,566	Test 3	
		Benign	Malign	
		992	421	

As the raw dataset consisted of mixed DICOM and JPEG images, DICOM mammograms were converted to JPEG format and anonymised before formatting all mammograms to have the same size of 2,346 × 2,969. The formatted dataset was labeled according to standardized breast imaging terminology by consensus into 6 BI-RADS categories before all images were labeled in compliance with the reports. Moreover, BI-RADS four images were categorized by risk into three subcategories: 4A, 4B, and 4C.

Concerning grouping, because the study is on diagnosis, although BI-RADS 6 represents a category of proven cancer cases, it may be integrated into a group with a very high-risk score (at least 95%), BI-RADS 5. Moreover, BI-RADS 4 could be included to form a so-called “malignant” group. By the same logic, BI-RADS 2 and BI-RADS 3 may be gathered to generate a “benign” group, while BI-RADS 0 and BI-RADS 1 should be left as they are. Hence, the dataset is regrouped into four groups in total. The details related to categories are given in Table 2.

An application for ethical approval was submitted to the Adana Alparslan Türkeş Science and Technology University Ethics Committee, with decision number 13-1, dated July 22, 2020. Although all images were anonymized before we got them, a written consent form from the patients was obtained.

Proposed hybrid DL model

In this study, a novel hybrid DL approach has been proposed for BI-RADS classification. The proposed model comprises three main phases: pre-processing, deep feature extraction, and classification, as illustrated in Fig. 1A. The pre-processing step contains ROI extraction with YOLOV8, resizing, left-flipping, and data augmentation. The left-flipping approach in this stage is also newly proposed in this work. In the deep feature extraction phase, a ConvNeXt-based inception architecture is employed to extract the features from breast images, as demonstrated in Figs. 1B and 1C. Detailed information about these stages is given in the following sub-sections.

Pre-processing

Mammogram interpretation is a very complex problem for several reasons. One of the most important aspects is the large number of parameters to be considered, as well as their interrelationships. When it comes to binary classification, into malignant and benign cases, or triple classification, where BI-RADS 1 cases are included, which are mainly based on separate analysis of masses, most of these parameters can, to some extent, be ignored. However, BI-RADS reporting cannot be reduced to these three groups, just as it cannot be reduced exclusively to the analysis of masses. Thus, for example, a mass in partial shadow, hidden by cysts, is less likely to be detected. For all the above reasons, the breast should be analyzed as a whole instead of focusing on marked masses within. That is to say, a larger number of parameters must be considered, and the preprocessing paradigm needs to be modified accordingly.

Pre-processing is essential not only to improve image quality but also to enhance the accuracy of the identification process (Tariq et al., 2021). Unrelated data is one of the biggest challenges of ML algorithms when training. Thus, image pre-processing, especially region of interest (ROI) extraction, is vital for removing unwanted areas of the image (Aslan, 2023). In this work, ROIs are automatically extracted from images using a pre-trained deep object detection method, YOLO-V8. Although it was reported that morphological image pre-processing approaches such as noise removal, histogram equalization, and morphological analysis could improve the image quality (Saber et al., 2021), this study uses none. The reason why none of them is applied is that the focus of the study is not to determine the ROI from the detected cancerous areas themselves, but from the entire breast tissue. Additionally, it is believed that possible pixel deviations do not have a major impact on the study. Hence, as shown in Fig. 1, pre-processing consists of four steps: ROI extraction, resizing, reorientation, and data augmentation. It should be highlighted that there is no manual correction during the pre-processing.

YOLO-V8 ROI extraction

For medical image analysis, ROI extraction is a vital pre-processing step, which provides a meaningful improvement in overall system efficiency. In most of the conducted studies, this ground-truth information is supplied from related datasets in which radiologists have already labeled ROIs manually, smoothing the way for their extraction. Although these on-target images give unarguably valuable information about the boundaries and locations of lesions, a key problem is regarding the way of their marking, which is mostly human-driven and prone to human errors. Besides, the judging criteria for the real BI-RADS model have to be grounded on examining the breast as a whole, not the lesions in part (Chougrad, Zouaki & Alheyane, 2018). For a clearer picture of the issue, the ROIs should include the full extent of breast tissue and its boundaries.

ROI extraction with high feature preservation is a challenging step toward a highly accurate analysis of mammograms with a rather complex composition. In addition to several conventional strategies, some DL methods have been employed to address this problem over the last few years. In DL methodology, ROIs extraction is predominantly categorized as an object detection problem.

Object detection has been one of the most popular topics in computer vision recently. The researchers focused on increasing the detection speed and accuracy of the objects. The YOLO model, which can learn generalizable representations of objects, was proposed in Redmon et al. (2015). YOLO is simple in that a single CNN concurrently predicts multiple bounding boxes and class probabilities, and presently optimizes detection performance. In this study, we extract the whole breast region as ROI using YOLO-V8 (Jocher, Chaurasia & Qiu, 2023) architecture, which is an updated version of YOLO introducing new features and improvements referring to speed, accuracy, and flexibility (Redmon et al., 2015). In this study, we employed a pre-trained YOLO-V8 model to extract the breast ROI (Nam, 2023), an example as shown in Fig. S1B.

Resizing

Resizing the images to appropriate dimensions ensures faster data processing. In the study, images were resized twice. Firstly, ROI images are resized to 512 × 512 pixels using bilinear interpolation to learn the proposed model in the following steps. The ROI images were resized again before training and testing the model to fit the model’s input. The proposed model’s input size was 224 × 224 pixels.

Left-flipping

Mammograms are images with left and right orientation, marked on the upper right or upper left corner, depending on the orientation of the breast. The picture in Fig. S1A shows a left-sided MLO mammogram with the stem-pointed labels at the right top.

To enhance data and standardize the learning process of the proposed model, all images are automatically left-flipped before the model’s training. The proposed left-flipping algorithm is shown as Algorithm 1. In the breast images, the vertical sum of pixel intensity values tends to depend more on the location of the breast. As shown in Fig. 2A, the vertical red line is at the threshold point. When the sum of the values located on the right side of this point is greater than the sum of the values on the left, the image is right-sided, and vice versa. After the algorithm determines that the image is not left-sided, it vertically flips the image at that point, turning it left. In Fig. 2B, the image is determined as the left side, and the algorithm returns the image unaltered.

Algorithm 1 Left-flipping algorithm.

 Data: img=inputImage	
 Result: leftImg	
  vertSum←[];	
  i←0;	
  isLeftSide←False;	
 for col∈img do	
   vertSum[i]=∑(col);	
   i=i+1	
 end	
 if ∑i=0len(vertSum/2)(vertSum[i])≥∑i=len(vertSum/2)len(vertSum)(vertSum[i]) then	
   isLeftSide=True	
 end	
 if not isLeftSide then	
   img=verticalFlip(img)	
 end	
  leftImg←img	

Figure 2 (A) Right-sided image, (B) left-sided image.

Data augmentation

One of the most important keys to setting a successful DL model is to collect as much data as possible for its training. The larger the dataset, the higher the success rate. However, especially in the medical sciences, collecting a large amount of labeled data, which supervised learning models require, is challenging due to the medical staff’s intense work hours.

Data augmentation (DA) has been widely adopted in DL tasks to address class imbalance, limited training data, and enhance model generalization. DA is a method based on processing the original data to generate new data for the training stage of the models. DA is commonly used in DL and computer vision studies and encompasses various techniques, including cropping, flipping, shifting, shearing, rotating, zooming in and out, filtering, and adding noise. Bringing out new data leads to scaling the dataset up, which, in turn, improves the prediction ability and robustness of the model (Shorten & Khoshgoftaar, 2019; Aslan, 2023). However, regarding mammography, several studies have highlighted certain inherent limitations, such as contentious anatomical plausibility and the removal of some essential diagnostic features (Chlap et al., 2021; Mikolajczyk & Grochowski, 2018). Consequently, anatomically incorrectly generated images or inadvertently altered or removed microcalcifications or tumours may lead to label preservation issues and reduced model reliability (Mikolajczyk & Grochowski, 2018; Shen et al., 2019; Oza et al., 2022a). Other reported limitations include a lack of semantic variability, distortion by domain-inappropriate augmentations such as brightness shifts or noise injection, and the inability to adequately represent rare pathological classes (Shen et al., 2019; Qin et al., 2019). Finally, there is a lack of standardized validation protocols to assess the real-world efficacy of augmentation strategies in mammography, making it challenging to translate augmented data into clinically trustworthy outcomes (Oza et al., 2022a; Blahová, Kostolný & Cimrák, 2025).

Depending on the study’s architecture, some subset of these methods can be used. In this study, rotation, width shift, height shift, shear, and zoom DA methods are employed with the following values: 20, 0.2, 0.2, 0.2, and 0.2, respectively. These methods are preferred because they do not affect the region(s) with lesions or tissue density. Figure 3 shows examples of DA images obtained by the proposed methods. The leftmost column represents the original images pre-processed up to augmentation.

Figure 3 (A–E) Example data outputs for augmentation techniques used in the study.

Deep feature extraction and classification

Image processing is one of the most popular areas of computer vision (CV) and artificial intelligence (AI). CNN architectures based on DL are widely used in various visual processing tasks, such as image classification, object detection, face recognition, and medical imaging (Kaya, Akat & Yildirim, 2025). Driven by the demand for improved object detection and recognition, extensive research on CNNs based on DL architectures in recent years has led to advances in several new architectures, which have quickly found applications in medical image analysis (Kaya, Akat & Yildirim, 2025). One such example is ConvNeXt architecture coined in Liu et al. (2022). Although inspired by the Vision Transformer architecture, ConvNeXt outperformed it for a short time. With its hybrid CNN design that combines channel-wise and spatial convolutions, ConvNeXt is especially well-suited for image classification tasks involving a wide range of complex and diverse features (Ahmed et al., 2023). Due to its novel architecture, which includes parallel paths across multiple layers, the model can efficiently capture different types of visual data. ConvNeXt can simultaneously process spatial and channel-wise data by combining both paths, increasing the representational power and improving the classification accuracy (Liu et al., 2022).

The study presents a novel architecture for deep neural networks (DNNs) that utilizes certain layers of the ConvNeXt architecture (Liu et al., 2022) as a feature extractor. It is concatenated with a single naive inception block, a module commonly used in inception architectures such as GoogLeNet (Szegedy et al., 2015; Saini & Susan, 2023). Using pre-trained models as feature extractors is a transfer learning (TL) approach where most layers of the main model are frozen (Kaya, Akat & Yildirim, 2025). We presented a TL approach in which we froze all layers to the stage 3 block 2 normalization layer of the pre-trained model ConvNeXt-tiny for the lower-level features and concatenated it with the layers of a randomly initialized naive inception module for the higher-level features, as shown in Fig. 1. Several randomly initialized higher layers are added as classifiers, namely the BatchNormalization, GlobalAveragePooling2D, Dropout, and dense layers, as shown in Fig. S3.

The decision to use ConvNeXt-tiny layers as a feature extractor was inspired by the promising classification results achieved by the ConvNeXt-tiny architecture in image recognition tasks. Using pre-trained models and starting blocks in the new combined architecture aims to enhance robustness and address class imbalance problems more effectively. We employed the pre-trained ConvNeXt-tiny model, originally trained on the ImageNet dataset. This approach facilitates the transfer of domain knowledge from the extensive ImageNet object dataset to the smaller mammography dataset in the lower layers while also enabling the model to acquire higher-level features that are specific to mammogram images. The naïve Inception block combined with a dense layer reduces computational costs (Saini & Susan, 2023). The inception block consists of three convolutional layers and a max pooling layer. Convolutional layers have filter sizes of 64, 128, and 32, and kernel sizes of 1, 3, and 5, respectively. The maxpooling layer has a 3×3 pool size, a stride of one, and the same padding.

Results

Experimental setup

This study performed experimental tests on a PC equipped with an Intel Core i7-10750 processor, an NVIDIA RTX 3060 GPU with 6 GB of VRAM, and 32 GB of RAM.

During the evaluation of the model, three scenarios, Test 1, Test 2, and Test 3, were employed, as detailed in ‘Numerical results’. A total of 2,566 images were split into a 70:10:20 hold-out validation set. In this approach, the dataset was randomly separated into 70% training, 10% validation, and 20% test sets. The model was trained with the training set and then evaluated with the validation set to assess its performance.

The model was trained in 100 epochs for the first phase of the training stage with the learning rate of 0.001 and 50 epochs for fine-tuning using the learning rate scheduler given in Fig. S2, which changes from 1e−5 to 5e−5, warming up in 110 epochs and learning rate exponential decay is 0.8. These parameter values were determined by empirical observations. The other learning parameters are as follows: the optimizer is Adam, the loss is categorical cross-entropy, and the batch size is 64.

Numerical results

Due to the complexity of the problem, the diagnosis was made through three experimental tests, presented as Test 1, Test 2, and Test 3. Test 1 involves the extraction of healthy breast mammograms through a binary classification, dividing the set into BI-RADS 1 and the remaining categories. Unlike other studies, neither the BI-RADS 1 group is part of the benign group, nor are the BI-RADS 0 cases discarded (Qian et al., 2024; He et al., 2023). In most studies conducted in the last decade, the BI-RADS 0 group has been underestimated and even neglected. However, as pointed out in Yi et al. (2022), effectively reducing unnecessary mammography recalls and biopsies without missing highly malignant tumors is quite challenging without including BI-RADS 0 cases and appropriate training to detect them. Test 2 is used to distinguish between BI-RADS 0 and other cases. Finally, Test 3 identifies malign (BI-RADS 4-6) and benign cases (BI-RADS 2-3) (Shen et al., 2019). For this purpose, the gathered dataset was reorganized into new 2-class test datasets. Both the first and second datasets comprise a total of 2,566 files, split into 1,795, 256, and 515 files for training, validation, and testing, respectively. As distinct from others, in the third experiment’s dataset, 988, 141, and 284 files are employed for training, validation, and testing, respectively. It should be emphasized that the third dataset includes neither the BI-RADS 0 nor the BI-RADS 1 cases.

In experimental tests, using the hold-out method, datasets were randomly selected, with 70% allocated for training and 10% for validation. The remaining 20% was used for testing. This ratio, used in various studies by Khalifa et al. (2021), Arafin, Billah & Issa (2024), covers standards for validation in deep learning (DL). The results were obtained from three tests via experiments repeated 10 times.

Table 3 reports average minimum, maximum, mean, and standard deviation (Stdev.) results for 10 experiment repetitions of three tests. As can be seen from the table, the mean values vary between 0.820775 and 0.862718. Although the best results were obtained for Test 1, the standard deviations of all tests do not exceed 0.012039, indicating that the data is highly clustered around the mean. From the same perspective, Test 3 had the worst performance. However, the standard deviation remains low, indicating that the model is robust and fits the data well.

Table 3 The obtained accuracy values of experimental tests.

	Test1	Test2	Test3	
Minimum	0.858252	0.842718	0.799296	
Maximum	0.869903	0.862136	0.841549	
Mean	0.862718	0.852816	0.820775	
Standard deviation	0.004259	0.005547	0.012039	

The diagnostic skill of the proposed method was examined on unseen data via a hold-out validation technique for various metrics such as precision, recall, specificity, and F1-score. Table 4 provides a closer look at results related to Test 1. Healthy cases are parsed with average mean precision, recall, specificity, and F1-score of 87.17%, 86.27%, 85.08%, and 86.53%, respectively. When comparing the parameters above for the formed groups, the performance of the second group is significantly better than that of the first. This means the model is better trained to find cases rated as “suspicious” or “needs deeper evaluation” than healthy ones. This problem stems not only from an unbalanced data set but also from the fact that patient age is an additional factor to be considered, especially in BI-RADS 1 scoring, having quite a large impact on tissue density (Ji et al., 2021). Numerical imbalance is better seen through the overall performance of the model presented by F1-score and the confusion matrix depicted in Fig. 4A. The false-positive rate (FPR) and false-negative rate (FNR), which are 0.094 and 0.42, respectively, also point to the imbalanced dataset.

Table 4 Precision, recall, specificity, and F1-score results of Test 1.

		Precision	Recall	Spesificity	F1-score	
0	Minimum	0.720670	0.807947	0.862637	0.769716	
	Maximum	0.744186	0.867550	0.879121	0.793846	
	Mean	0.730719	0.842384	0.871154	0.782490	
	Standard deviation	0.007806	0.015332	0.005950	0.007478	
1	Minimum	0.916905	0.862637	0.807947	0.896159	
	Maximum	0.940120	0.879121	0.867550	0.905233	
	Mean	0.930244	0.871154	0.842384	0.899700	
	Standard deviation	0.006060	0.005950	0.015332	0.003115	
Avg.	Minimum	0.863552	0.858252	0.828815	0.860115	
	Maximum	0.878572	0.869903	0.866109	0.872384	
	Mean	0.871743	0.862718	0.850820	0.865334	
	Standard deviation	0.004678	0.004259	0.010036	0.004233	

Figure 4 Average confusion matrix on ten runs for: (A) Test 1, (B) Test 2, and (C) Test 3.

The numerical results obtained for Test 2 are summarized in Table 5. Test 2 achieved mean precision, recall, specificity, and F1-score of 85.72%, 86.27%, 62.10%, and 85.48%, respectively, which are the worst results obtained in the study. A relatively low specificity indicates that the model’s “true BI-RADS 0 rate” is also low. Specifically, due to an imbalance in groups of the dataset, it is still vague for the model what the BI-RADS 0 is, and any predicted BI-RADS 0 has a chance to be any other BI-RADS, which also comes to the attention analyzing the average confusion matrix illustrated in Fig. 4B. It should be noted that, unlike other articles, the average confusion matrices in this study include all values along with their standard deviations.

Table 5 Precision, recall, specificity, and F1-score results of Test 2.

		Precision	Recall	Specificity	F1-score	
0	Minimum	0.500000	0.518519	0.900922	0.509091	
	Maximum	0.560976	0.629630	0.917051	0.582857	
	Mean	0.529780	0.567901	0.905991	0.547868	
	Standard deviation	0.016384	0.035353	0.004811	0.023075	
1	Minimum	0.909513	0.900922	0.518519	0.906358	
	Maximum	0.928741	0.917051	0.629630	0.918108	
	Mean	0.918312	0.905991	0.567901	0.912089	
	Standard deviation	0.006025	0.004811	0.035353	0.003195	
Avg.	Minimum	0.845104	0.842718	0.579026	0.843876	
	Maximum	0.868001	0.862136	0.672299	0.863444	
	Mean	0.857203	0.852816	0.621076	0.854803	
	Standard deviation	0.007123	0.005547	0.029522	0.006061	

Table 6 details the results achieved for Test 3 related to precision, recall, specificity, and F1-score. Their mean values are obtained as 81.81%, 82.07%, 73.33%, and 81.89%, respectively. The results from the table highlight that the dataset is more balanced, but for better results, the dataset should be a little more comprehensive. The same is apparent from the average confusion matrix for 10 repetitions in Fig. 4C.

Table 6 Precision, recall, specificity, and F1-score results of Test 3.

		Precision	Recall	Specificity	F1-score	
0	Minimum	0.844660	0.864322	0.623529	0.859259	
	Maximum	0.877451	0.904523	0.705882	0.888337	
	Mean	0.862211	0.885930	0.668235	0.873839	
	Standard deviation	0.010335	0.012920	0.028721	0.008624	
1	Minimum	0.679487	0.623529	0.864322	0.650307	
	Max	0.750000	0.705882	0.904523	0.727273	
	Mean	0.714973	0.668235	0.885930	0.690447	
	Standard deviation	0.024010	0.028721	0.012920	0.021497	
Avg.	Minimum	0.795225	0.799296	0.698605	0.796721	
	Max	0.839305	0.841549	0.763831	0.840131	
	Mean	0.818144	0.820775	0.733390	0.818951	
	Standard deviation	0.012286	0.012039	0.020225	0.012148	

To compare the proposed model with the baseline models and conduct an ablation study, examining the use of the Inception module, we have performed experiments. To this end, we employed EfficientNetB0 and ConvNextTiny in each experimental test scenario, achieving the results presented in Table 7. As shown in the table, the baseline models yielded similar results. The proposed model, Mam-Incept-Net, attained better results. These results demonstrate that incorporating Inception modules into the proposed model enhances its generalization ability.

Table 7 Comparison with baseline models.

	EfficientNet	ConvNext	Mam-Incept-Net	
Test1	0.844660	0.840776	0.869903	
Test2	0.850485	0.825242	0.862136	
Test3	0.802816	0.809859	0.841549	

To make a statistical comparison of the models, Friedman’s chi-square and Nemenyi tests were performed using the data in Table 7. The p-value was calculated as 0.0969 in the Friedman chi-square test. Considering the significance level of α is 0.10, it can be evaluated that there is a statistically significant difference between the methods. As shown in Table 8, there is no significant difference between methods 1 and 2; however, significant differences are observed between methods 1 and 3, and between methods 2 and 3. However, it is recommended that statistical tests be performed with more data to achieve a more comprehensive analysis.

Table 8 The statistical results of the Nemenyi test.

	1	2	3	
1	1.000000	0.912237	0.231725	
2	0.912237	1.000000	0.102484	
3	0.231725	0.102484	1.000000	

Discussion

This article presents a new model for classifying mammograms into four BI-RADS categories. The model is performed on the private dataset, with no excluded BI-RADS category. Regarding the regrouping of images labeled by their own BI-RADS, instead of seven, the dataset is reorganized into four tailored sub-datasets. The Benign group is formed by combining BI-RADS 2 and BI-RADS 3, while the Malign group comprises BI-RADS 4, BI-RADS 5, and BI-RADS 6. The way of creating a Malignant group shares some similarities with Qian et al. (2024), Malebary & Hashmi (2021). However, two important differences related to the regrouping process as a whole should be underlined. The first is about the BI-RADS 1 group, considered independent in the study. This consideration is based on medical grounds, as the BI-RADS 1 group is solely the so-called “healthy” group and needs no additional medical treatment. Another vital difference refers to the BI-RADS 0 group, which was predominantly discarded in previous studies.

From the medical point of view, this is a group that is hardly diagnosed and is often confused, frequently seen in young patients whose tumours and metastasis seem to be more aggressive. Hence, neglecting or attaching BI-RADS 0 to the other groups may lead to misdiagnosis or overdiagnosis. Hence, in our opinion, BI-RADS 0 needs to be classified separately. Additionally, the way of creation of the B group is in line with the studies conducted in Aly et al. (2021), Dhungel, Carneiro & Bradley (2017). Nevertheless, they don’t use all BI-RADS groups.

Apart from the group tailoring, one of the marked differences is in automatically extracted ROIs containing the breast image as a whole instead of manually labeled ones. As BI-RADS grading is a multi-variable problem, including asymmetry, architectural distortion, calcification, mass, and associated features, it is hardly believed that whole breast ROIs could be helpful to avoid information loss. Moreover, the automatically extracted ROIs could reduce human errors during the labeling.

Classification is provided by ConvNeXt, a powerful DL model that performs exceptionally well on image datasets during the learning process. In the last few years, ConvNeXt has been used for various medical purposes, especially in radiology classification. In several studies, such as those conducted in Al-Mansour et al. (2023), Huynh, Tran & Tran (2023), ConvNeXt-based models were reported to be very suited for mammogram classification. However, in the studies above, ConvNeXt is used as a binary classifier working on just two of five sub-problems and sorting out manually extracted ROIs collected from the public datasets into benign and malignant classes or as a detector for potential BC lesions, generating ROIs automatically.

Conclusion

Thus far, mammogram classification has been predominantly based on examining one or two of several decisive parameters or discarding one or more BI-RADS categories. Mostly, manually marked and extracted ROIs have been pre-processed and classified into two or three categories. Due to the complex and multi-parametric nature of the issue, such approaches have fallen behind the real BI-RADS scoring.

The proposed model represents a more realistic solution for mammogram classification based on BI-RADS assessment categories, which are used as the sole criterion for describing the results of breast exams. This approach also examines the problem of BI-RADS scoring in its entirety. For this purpose, the automatically extracted ROIs containing the breast with its boundaries are judged and classified with a novel ConvNeXt-based fusion model, Mam-Incept-NET.

The scoring is based on three experimental tests, with results ranging from 82.08% to 86.27%. Although the obtained experimental results appear to be slightly higher than the analogies at first glance, these results are based on a more comprehensive examination. The obtained results clearly show that the variety of sub-problems related to the BI-RADS 0 group has negatively impacted the model’s performance. Despite the indicated difficulties, we are convinced that the proposed system will significantly aid the radiologist in the early diagnosis of breast cancer.

Numerical results can be improved by augmenting raw data and reducing the quantitative discrepancy between benign and malignant datasets. It should be noted that collecting properly labeled malignant images is a challenging task. Moreover, augmentation strategies for mammography are limited, making it difficult to translate augmented data into clinically reliable outcomes.

After fortifying it with the previous examination reports, the model would provide a comprehensive picture of BI-RADS grading, as the differences between BI-RADS 2-3 or among BI-RADS 4, 5, and 6 are obtained by considering the previous mammography, MRI, and biopsy reports. They also play an essential role in distinguishing between mammograms classified as BI-RADS 4A, 4B, and 4C, as the differences are slight and may obstruct the correct diagnosis. Harmonization between the grading and examination systems, taking into account prior reports, will be a topic of our future work.

Supplemental Information

Supplemental Information 1 Mam-Incept-NET model’s Python code.

Supplemental Information 2 BI-RADS Assessment Categories.

Supplemental Information 3 Examples of types of noise and ROI detection in an image.

Supplemental Information 4 Learning rate schedule for fine-tuning.

Supplemental Information 5 The model diagram of the proposed approach.

Additional Information and Declarations

Competing Interests

The authors declare that they have no competing interests.

Author Contributions

Amira Tandirovic Gursel conceived and designed the experiments, analyzed the data, prepared figures and/or tables, authored or reviewed drafts of the article, and approved the final draft.

Yasin Kaya conceived and designed the experiments, performed the experiments, performed the computation work, prepared figures and/or tables, authored or reviewed drafts of the article, and approved the final draft.

Ethics

The following information was supplied relating to ethical approvals (i.e., approving body and any reference numbers):

An application for ethical approval was made to the Adana Alparslan Türkeş Science and Technology University Ethics Committee, with decision number 13-1, dated 22.07.2020.

We do not require a consent form because the data is anonymized.

Data Availability

The following information was supplied regarding data availability:

The dataset is available at GitHub and Zenodo:

- https://github.com/yasinkaya1/Mam-Incept-NET/tree/main/dataset.

- Tandirovic Gursel, A., & KAYA, Y. (2025). Mammogram images with BIRADS [Data set]. Zenodo. https://doi.org/10.5281/zenodo.15730564.

The code of the proposed model is available in the Supplemental File.

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
