# Peer review of "Mam-Incept-Net: a novel inception model for precise interpretation of mammography images"

_PeerJ Computer Science, doi:10.7717/peerj-cs.3149_

## Round 0.1 · original submission · Major Revisions

**Language Note:** The review process has identified that the English language must be improved. PeerJ can provide language editing services - please contact us at [email protected] for pricing (be sure to provide your manuscript number and title). Alternatively, you should make your own arrangements to improve the language quality and provide details in your response letter. – PeerJ Staff

Reviewer 1 ·

Basic reporting

Review:
Mam-Incept-Net: A novel inception model for precise interpretation of mammography images

Major Comments
1. Dataset Description and Experimental Clarity
i. Clearly state the number of images per BI-RADS category and address any class imbalance issues.
ii. Explain how labels were assigned: single radiologist, majority vote, or consensus?
iii. Describe how data was split for training, validation, and testing. Was cross-validation used?
iv. Report average performance across splits and include standard deviation or confidence intervals.
2. Architecture and Model Details
i. Describe the architecture more precisely. What are the exact layers and parameters in the inception module?
ii. Clarify whether ConvNeXt was pretrained or trained from scratch. If pretrained, mention the dataset.
iii. Explain how the two modules (ConvNeXt and inception) are fused and why ?
iv. Include an ablation study to show the added value of the inception module over using ConvNeXt alone.
3. ROI Detection
i. Provide details on how well YOLO-V8 performed in detecting regions of interest.
ii. Clarify if there was any manual correction of ROIs or filtering of failed detections.
iii. Consider showing example detections or providing recall/precision metrics for the ROI step.
iv. Please consider adding a paragraph or even a subsection discussing the limitations of traditional data augmentation techniques (e.g., flipping, rotation, scaling) in the context of mammography.
4. Evaluation of Metrics and Result Interpretation
i. In addition to accuracy, report F1-score, per-class accuracy, and confusion matrix to assess class-level performance.
ii. Compare Mam-Incept-Net to at least one baseline.
iii. Clarify the context of the statement that “accuracy dropped compared to previous studies”—cite specific studies and numbers.
iv. Ensure that all tables explain what each metric means and what data it comes from.
5. Figures and Tables
i. Ensure all figures have descriptive captions.
ii. In any graphs, ensure axes are labeled and include units.
iii. Where applicable, include error bars or performance variability in tables.

Experimental design

The experimental design involves training the proposed Mam-Incept-Net on a real-world dataset of 2,571 mammograms from two hospitals, covering all BI-RADS categories. The model uses YOLO-V8 for ROI extraction and combines ConvNeXt with a custom inception module. Performance is evaluated across three runs using accuracy and F1-score to ensure robustness and consistency.

Validity of the findings

The findings of the study appear valid, as the proposed Mam-Incept-Net was evaluated on a real-world, diverse dataset with all BI-RADS categories represented, enhancing clinical relevance. The use of multiple experimental runs adds reliability, and consistent performance metrics (accuracy and F1-score) support the model’s robustness. However, the absence of detailed class-wise evaluation and external validation limits full generalizability, which should be addressed in future work.

Additional comments

Minor Comments
1. Writing and Language
i. Correct minor typos throughout the whole manuscript.
ii. Standardize the formatting of BI-RADS and avoid switching between “BIRADS” and “BI-RADS”.
2. Clarity and Consistency
i. Define acronyms (e.g., “PM dataset” in the conclusion is not explained).
ii. Explain “3 experimental tests” more clearly—is it 3 folds or 3 separate runs?
iii. Ensure metric definitions like accuracy, F1, etc. are consistent throughout the manuscript.
3. Related Work and Citations
i. Please consider citing the following recent and relevant works to strengthen the literature review and position your contributions more clearly in the context of current deep learning approaches for mammogram analysis: Enhancing the Quality and Authenticity of Synthetic Mammogram Images for Improved Breast Cancer Detection, Reliable Breast Cancer Diagnosis with Deep Learning: DCGAN‐Driven Mammogram Synthesis and Validity Assessment, Optimizing Breast Cancer Detection With an Ensemble Deep Learning Approach.
ii. Cite directly when mentioning previous studies with higher accuracy to support your comparison.
iii. Ensure all references mentioned in-text are fully included in the reference list.

·

Basic reporting

Needs broader referencing to recent models in mammography AI.
Dataset details are vague.
Reasonably clear but requires edits for better readability.

Experimental design

Research question is defined but needs to be more specifically stated.
Architecture and training parameters could use more details.

Validity of the findings

Needs statistical and error analysis.
Anything you claim, back it up by specifics and more details.

Additional comments

This article addresses an important challenge in computer-aided diagnosis which is automated and inclusive classification of mammograms using BI-RADS categories. You have used ConvNeXt and inception modules for multi-scale feature extraction and applied it to a clinically relevant task.
You could use more methodological details, and adequate baseline comparison for better impact. Thank you for this study, enjoyed reading it. Refer peer review comments below .

Abstract:
1. The abstract introduces the model but lacks a clear statement on the information about novel approach.
2. Phrases like “slight drop in accuracy” are ambiguous and should be avoided or backed up with data. You may need to quantify this directly and explain its implications clearly.
3. Clarify what is “authentic dataset”. Is it private or public? Include name and source as well.

Introduction:
1. This section correctly identifies the limitations of current mammography interpretation, but it doesn’t clearly specify what gap this work addresses.
2. For BI-RADS and deep learning – can you add reference for recent state-of-the-art research. Ex. EfficientNet, DenseNet, or ViT applied to mammography.
3. The reason for integration of ConvNeXt is not well explained. Can you add more information why ConvNeXt is used for low-level and inception for high-level features.
4. Objective should be stated in clearer terms, right now there is no clear goal / objective for study. Also include who can get benefit from this research.

Methodology:
1. Can you include figure that shows how ConvNeXt and inception modules are connected.
2. Can you describe configuration of the inception module ex. number of filters, kernel sizes, and structure of each path in the module. These details are important.
3. Did you do any data preprocessing – how ROIs were extracted, segmentation step if used, how did annotations play role in it – were they provided or inferred?
4. It is not very clear what was training procedure – there are details missing ex. optimizer used, learning rate, number of epochs, and data augmentation techniques etc.
5. This is critical in medical imaging where data leakage is a risk. How did you split the images patient-wise or randomly?

Experimental Setups:
1. This section mentions “authentic dataset” – could you add more details ex. is this clinical dataset, how many patients/images per class
2. Any information on class imbalance and generalizability – can you show breakdown of how many samples fall into each BI-RADS class?
3. The accuracy is reported but it is unclear what baselines this model is compared against. Add comparioson against 1-2 baseline models.
4. What was the validation protocol - cross-validation or hold-out split? Was any stratification applied?

Results & Discussion:
1. Accuracy is good but Include confusion matrices, AUC, F1-scores per class, especially in a medical imaging where false negatives are critical.
2. How did you validate if performance differences are statistically significant. Can you add t-test, bootstrap CI etc.
3. There is no ablation study done here – can you add that comparing ConvNeXt-only, Inception-only, and Mam-Incept-NET hybrid.
4. Can you use Grad-CAM or SHAP to explain model decisions and where model fails.

Conclusion:
1. You mention drop in accuracy is a trade-off for inclusivity, even though it makes sense can you quantify how inclusivity affected class-wise performance?
2. Are there limitations in this study ex. dataset size, class imbalance, and lack of external validation.
3. Can you mention specific future work directions.
4. Any thoughts how the model could be integrated into current radiology workflows or CAD systems.

---

## Round 0.2 · accepted · Accept

Both reviewers are satisfied with the revised manuscript, and I am happy to accept this manuscript for publication in PeerJ Computer Science.

Reviewer 1 ·

Basic reporting

No more comments

Experimental design

No more comments

Validity of the findings

No more comments

Additional comments

No more comments

·

Basic reporting

This version looks better, thanks for addressing all review comments.

Experimental design

This version looks better, thanks for addressing all review comments.

Validity of the findings

This version looks better, thanks for addressing all review comments.

Additional comments

This version looks better, thanks for addressing all review comments.